# Self-predictive Mamba: Improving Multi-agent Reinforcement Learning with Self-predictive Encoding

## Abstract

In multi-agent reinforcement learning (MARL), agents must collaborate to achieve team goals while only having access to limited local observations. This partial observability, coupled with the dynamic presence of other agents, renders the environment non-stationary for each agent, complicating the policy training. A critical challenge in this setting is the efficient utilization of historical information for decision-making. Building on the hypothesis that self-predictive features can improve policy learning, we introduce the self-predictive Mamba, a novel framework that integrates the Mamba model with self-predictive representation learning for decentralized policy optimization. Self-predictive Mamba leverages a unique policy architecture where the Mamba model is trained to predict future observations, aiding in more stable and informed decision-making. Substantial experiments demonstrate that self-predictive Mamba significantly outperforms the widely used recurrent neural network (RNN)-based MARL policies and surpasses those naively employing the Mamba model.

## 1 Introduction

Multi-agent reinforcement learning (MARL) has gained considerable attention due to its successes in domains such as real-time strategy games (Ye et al., 2020), autonomous driving (Zhou et al., 2021), and robot swarm navigation (Hüttenrauch et al., 2019). However, one primary challenge in MARL lies in the inherent non-stationarity of the environment. As multiple agents continuously update their policies, the dynamics of environmental transitions remain in flux from the perspective of any single agent. Additionally, agents in decentralized settings have only limited, local observations, which can hinder effective policy training.

While assuming global access to the environment could mitigate this issue, the associated communication overhead is often prohibitive, making such approaches impractical. Consequently, state-of-the-art (SOTA) methods frequently employ RNNs to extract spatial-temporal features for decision-making in multi-agent settings (Rashid et al., 2020; Yu et al., 2022). Despite their utility, standard RNN architectures struggle with the complexity of MARL environments, leading to suboptimal policy performance.

Recently, Transformers (Vaswani et al., 2017) have gained prominence in RL for their strength in sequential reasoning (Zambaldi et al., 2019; Sridhar et al., 2023; Klissarov et al., 2023; Zhang et al., 2023). However, their application to decision-making in the context of MARL remains relatively limited. Transformers are notoriously data-hungry, often requiring large datasets for effective training. Pre-training with large language models (LLMs) has been explored to alleviate this issue (Shi et al., 2024; Radford et al., 2019; Nottingham et al., 2023), but it only offers partial relief while adding a significant computational overhead and slower inference times. Moreover, Transformers tend to underperform in non-stationary environments, which limits their applicability to specific RL paradigms such as offline and in-context learning (Chen et al., 2021; Logeswaran et al., 2023; Jiang et al., 2023; Zhang et al., 2024). These limitations are further amplified in multi-agent scenarios, where complex inter-agent dynamics exist. Although modeling MARL as a multi-agent sequential decision-making process (Wen et al., 2022) offers some mitigation, the stringent assumptions required by this approach limit its general applicability.

State space models (SSMs) (Zadeh & Desoer, 2008), on the other hand, offer a promising alternative to RNNs by efficiently utilizing historical information for decision-making. SSMs have demonstrated superior performance in deep representation learning and sequential reasoning tasks, often outperforming well-established models such as Transformers and LSTMs (Graves, 2012). Unlike Transformers, which scale quadratically with the sequence length, SSMs offer much faster inference speeds and scale linearly, making them computationally more efficient and scalable in practice (Gu & Dao, 2023). Furthermore, SSMs can be directly applied to iterative decision-making processes, bypassing many of the limitations associated with Transformers. However, in general, SSMs remain difficult to train, especially for decision-making in non-stationary MARL settings, as they suffer from relatively low data efficiency.

Self-supervised representation learning has recently been introduced in RL to create auxiliary predicting objectives, which can enhance the feature representation learning and task performance (Schwarzer et al., 2020; Fang & Stachenfeld, 2024). In action decision-making based on historical sequences, self-supervised representation learning is typically used to learn world models by predicting future observations. In partially observable MARL tasks, it is natural to hypothesize that learning predictable representations can contribute to policy training.

Motivated by the success of SSMs in sequence modeling and the effectiveness of self-supervised learning in world models, we propose self-predictive Mamba (also abbreviated as SP-Mamba), a novel framework for action inference in decentralized MARL settings. Self-predictive Mamba relies on the Mamba model (Gu & Dao, 2023) to extract spatial-temporal features from agents' historical observations. To ensure consistency and predictability in these features, we introduce a multi-layer perceptron variational auto-encoder (MLP-VAE), which aggregates dense representations from raw observations and a decoder that predicts the next encoder output based on the model's output. Unlike typical world model training, the MLP-VAE is trained jointly with the policy, focusing solely on reward maximization without input reconstruction or representation learning. Our experimental results show that self-predictive Mamba significantly outperforms RNN-based policies on challenging tasks from the StarCraft II multi-agent challenge (SMAC) (Samvelyan et al., 2019), while also providing substantial improvements over the direct application of Mamba for action decision-making.

## 2 BACKGROUND

### 2.1 PARTIALLY OBSERVABLE MARKOV DECISION PROCESS

In MARL, the goal is to learn policies for multiple agents that interact within a shared environment, aiming to maximize a scalar reward signal. This problem is often modeled as a partially observable Markov decision process (POMDP), represented by the tuple $G = \langle S, U, P, R, O, N, \gamma \rangle$, where $s \in S$ represents the global state about the entire multi-agent system and the environment; $u_i \in U$ is the action taken by the $i$-th agent forming the joint action $\boldsymbol{u} = [u_1 \ u_2 \ \cdots \ u_N] \in U^N$; $P(s'|s, \boldsymbol{u})$ : $S \times U^N \times S \to [0, 1]$ is the state transition function; $R$ is the reward function $r = R(s, \boldsymbol{u})$ that gives a scalar reward signal according to the global state $s$ and the joint action $\boldsymbol{a}$; $O$ describes the observation function that generates observation $o_i$ for the $i$-th agent; $N$ is the total number of agents; and $\gamma$ is the discount factor in calculating returns. At each time step, each agent $i \in N$ receives an observation $o_i \in O$ and selects an action $u_i$, resulting in a joint action $\boldsymbol{u} = [u_1 \ u_2 \ \cdots \ u_N] \in U^N$. The global states then transits to $s'$ according to $\boldsymbol{u}$ and the reward $r$ is given. In fully collaborative settings, the reward is shared by the entire multi-agent system. The objective of policy training is to learn a policy $\pi(u_i|o_i)$ to maximize the total expected return $L(\pi) = \mathbb{E}_\pi[\sum_{t=0}^{\infty} \gamma^t r(s_t, \boldsymbol{u}_t)]$.

### 2.2 MAMBA MODEL

The Mamba model is a bunch of SSMs that include a selection mechanism to efficiently handle sequential data (Gu & Dao, 2023). At each time step $t$, the Mamba model takes an input $x_t$, updates a latent state $h_t$, and produces an output $y_t$. The discrete form of the Mamba model is defined as

$$h_t = \bar{\boldsymbol{A}}_t h_{t-1} + \bar{\boldsymbol{B}}_t x_t \tag{1a}$$

$$y_t = \boldsymbol{C}_t h_t + D x_t \tag{1b}$$

Here, $\bar{\boldsymbol{A}}_t = f_A(\Delta_t, \boldsymbol{A}_t)$ and $\bar{\boldsymbol{B}}_t = f_B(\Delta_t, \boldsymbol{A}_t, \boldsymbol{B}_t)$, $\boldsymbol{C}_t$ are discrete model parameters, $D$ is a learnable weight that controls the contribution of the original input $x_t$ through a shortcut connection.

In the standard Mamba implementation, discretization follows the zero-order hold (ZOH) method

$$\bar{\boldsymbol{A}}_t = \exp(\Delta_t \boldsymbol{A}_t) \tag{2a}$$

$$\bar{\boldsymbol{B}}_t = (\Delta_t \boldsymbol{A}_t)^{-1}(\exp(\Delta_t \boldsymbol{A}_t) - I)\Delta_t \boldsymbol{B}_t \tag{2b}$$

While $\boldsymbol{A}_t$ is initialized and updated independently, the Mamba model employs a selection mechanism, using linear projections to compute the matrices $\boldsymbol{B}_t = f_B(x_t)$, $\boldsymbol{C}_t = f_C(x_t)$, and $\Delta_t = f_\Delta(x_t)$. To operate over an input sequence $x$ of batch size $B$ with $L$ channels, the SSM in equation 1 is applied independently to each channel with model parameters $\boldsymbol{A}_t \in \mathbb{R}^{B \times L \times N \times N}$, $\boldsymbol{B}_t \in \mathbb{R}^{B \times L \times N \times 1}$, $\boldsymbol{C}_t \in \mathbb{R}^{B \times L \times 1 \times N}$ and $h_t \in \mathbb{R}^{B \times L \times N \times 1}$, the projections are defined as follows

$$
\begin{aligned}
f_B(x) &= \text{Linear}_N(x) \\
f_C(x) &= \text{Linear}_N(x) \\
f_\Delta(x) &= \text{Broadcast}_L(\text{Linear}_1(x))
\end{aligned}
\tag{3}
$$

where the operator $\text{Linear}_N(\cdot)$ projects to an $N$-dimensional space, and $\text{Broadcast}_L(\cdot)$ broadcasts a scalar to an $L$-dimensional space. The term $\Delta_t$ acts analogously to the gating mechanism $g_t$ in $h_t = (1 - g_t)h_{t-1} + g_t x_t$, controlling whether to reset the state based on the current input. Specifically, $\Delta_t$ decides whether to "select" and incorporate the current input $x_t$ while forgetting the previous state, or to retain the current state and ignore the input. Meanwhile, $\boldsymbol{B}_t$ and $\boldsymbol{C}_t$ serve as more fine-grained controllers, determining how $x_t$ influences $h_t$ and how $h_t$ contributes to $y_t$.

## 3 THE SELF-PREDICTIVE MAMBA MODEL

We focus on improving the learning stability and efficiency of decentralized RL policies in multi-agent collaboration tasks, which are often modeled as a POMDP. As the self-predictive Mamba model is designed for individual agent decision-making, we omit agent-specific indices in our description whenever clear from context. For example, the observation of the $i$-th agent at time $t$ is denoted as $o_t$, corresponds to $o_t^i$ in full notation. Additionally, for simplicity, variables related to a single agent or time step are denoted with regular symbols, while those associated with multiple agents or time steps are represented using bold symbols. For instance, the observation sequence of the $i$-th agent from time $t$ to $t + k$ is denoted as $\boldsymbol{o}_{t:t+k} = [o_t^\top, o_{t+1}^\top, \ldots, o_{t+k}^\top]$.

**Model components** Self-predictive Mamba extends the Mamba model to decentralized multi-agent decision-making settings. While the standard implementation of Mamba processes 1D sequences in parallel, we modify it for sequential decision-making on a step-by-step basis. The self-predictive Mamba consists of 5 components

| | | |
|---|---|---|
| Encoder | $x_t = \text{ENC}_\phi(\boldsymbol{o}_{t-k+1:t})$ | (4a) |
| Latent model | $h_t = \text{SSM}_\phi(h_{t-1}, x_t)$ | (4b) |
| Output projection | $y_t = \text{PROJ}_\phi(h_t, x_t)$ | (4c) |
| Decision maker | $u_t = \text{ACT}_\phi(y_t)$ | (4d) |
| Transition decoder | $\hat{z}_{t+1} = \text{DEC}_\phi(y_t, u_t)$ | (4e) |

Here, $\phi$ represents the learnable parameters of the self-predictive Mamba model. Each component is implemented as either neural networks or probability density variables for categorical distributions. The overall structure of self-predictive Mamba is shown in Figure 1.

**Input sequence encoding** Self-predictive Mamba employs an MLP-VAE in equation 4a to encode raw input data. To reduce redundancy in the raw observations, the observation $o_t$ is firstly encoded into a latent representation $z_t \in \mathbb{R}^{1 \times d\epsilon}$. Here, $d$ denotes the dimension of the model's output $y_t$, and $\epsilon$ is the model's expansion coefficient. The historical sequence of encoded observations $\boldsymbol{z}_{t-k+1:t}$ is then aggregated into $x_t \in \mathbb{R}^{1 \times d\epsilon}$ through a convolution neural network (CNN) The encoder is implemented using a single-layer MLP and a CNN.

**Mamba latent model** The latent model in equation 4b is implemented as a Mamba model. In this case, the $\text{SSM}_\phi$ in equation 4b denotes equation 1a. The latent model updates the hidden state $h_{t-1} \in \mathbb{R}^{1 \times d\epsilon \times 1}$ to $h_t$ utilizing the encoded observation $x_t$. All linear projections for calculating $\bar{\boldsymbol{A}}_t$ and $\bar{\boldsymbol{B}}_t$ are implemented as single-layer MLPs.

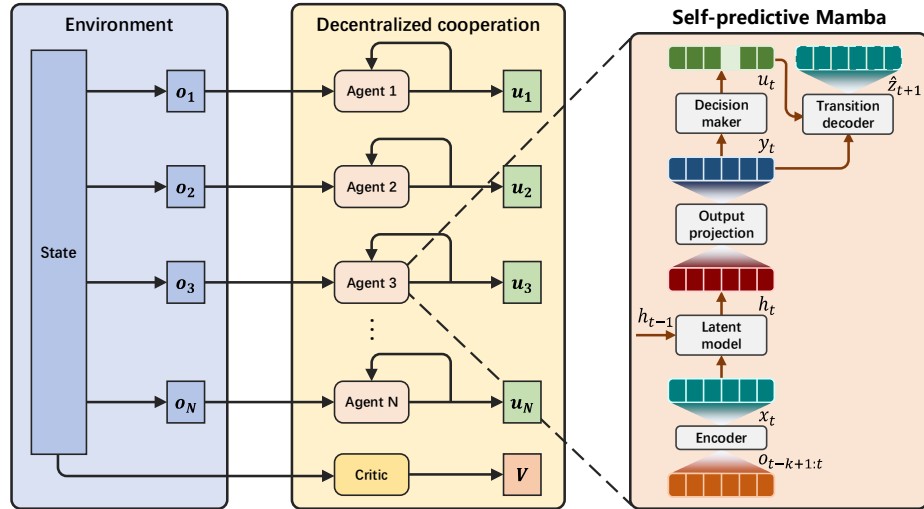

Figure 1: The structure diagram of the proposed self-predictive Mamba algorithm, where each agent is implemented as a self-predictive Mamba model presented in the right part of the figure. Self-predictive Mamba recurrently updates the hidden state $h_t$ from the encoded input $z_t$. For computing efficiency, the length of $\boldsymbol{o}_{t-k+1:t}$ is fixed to $k$. Each time $o_t$ is stored in, $o_{t-k}$ is popped out.

**Nonlinear output projection** The output projection in equation 4c computes $y_t$ through equation 1b. It applies a nonlinear transformation using a coefficient that is parameterized by $x_t$ and activated by SiLU (Elfwing et al., 2018)

$$y_t = \text{NORM}\big(y_t \odot \text{SiLU}[\text{MLP}_\phi(x_t)]\big) \tag{5}$$

where $\text{NORM}(\cdot)$ refers to the layer normalization and $\odot$ is the Hadamard production. The linear projection used in computing $C_t$ is implemented as a single-layer MLP.

**Categorical activation** The decision maker, represented by equation 4d, outputs a discrete categorical distribution over the available actions parameterized by the input variables. The action is selected by sampling once from this categorical distribution.

**Transition decoder** The transition decoder, implemented similarly to the encoder, is a single-layer MLP. Unlike common world model frameworks that regularize the latent representation towards a prior distribution, we train the decoder solely to predict the next encoder output. The rationale is that the model's output will be automatically regularized through end-to-end policy training, eliminating the need for explicit regularization toward a prior.

## 4   SELF-PREDICTIVE MAMBA FOR POLICY LEARNING

In this work, we implement self-predictive Mamba into MAPPO (Yu et al., 2022) and illustrate how the resultant actor and critic will be trained next. While we focus on MAPPO in our simulations, it is worth noting that self-predictive Mamba is designed as a general policy module, making it easily integrable into other policy-based algorithms that adopt distributed execution as well.

**Experience buffer** A buffer storing the past experiences about executing the policy is maintained to train the policy. The buffer contains sequences of agents' observations $\boldsymbol{o}_{1:N,0:T-1}$, global states $\boldsymbol{s}_{0:T-1}$, actions $\boldsymbol{u}_{1:N,0:T-1}$ along with the action log probabilities, the team rewards $\boldsymbol{r}_{0:T-1}$ and the next observations $\boldsymbol{o}_{1:N,1:T}$, where $T$ is the maximum step per episode and $N$ is the total number of agents. We randomly shuffle the data in the whole buffer and pack them into one single batch to ensure the stability of training. At the end of each training phase, the buffer is cleared in preparation for storing new experience data.

**Actor learning** The actor $u_t = \pi_\phi(o_t)$ is realized using the self-predictive Mamba policy designed in Section 3. We enable the experience data sharing among all the agents, and therefore, the experi-

Table 1: Average win rate for the self-predictive Mamba against Mamba, GRU, RODE (Wang et al., 2021) and QMIX (Rashid et al., 2020). All the results are presented in the 'average±variance' form. The columns with notation * represent the results obtained by the corresponding method using the same runtime with self-predictive Mamba. We report the runtime in Appendix B.

| Task | Self-predictive Mamba | Mamba | GRU | *RODE | *QMIX |
|---|---|---|---|---|---|
| 3s_vs_5z | **89.5±7.9** | 83.2±29.7 | 23.4±42.7 | 64.2±23.8 | 41.7±43.4 |
| 5m_vs_6m | **41.0±24.7** | 36.7±11.4 | 23.0±31.3 | 32.4±6.6 | 21.8±6.4 |
| 8m_vs_9m | **95.6±0.5** | 88.2±3.6 | 78.6±8.9 | 62.8±4.5 | 43.0±16.3 |
| 6h_vs_8z | **47.3±24.5** | 25.5±7.3 | 24.9±7.9 | 4.6±8.0 | 12.6±11.8 |
| 3s5z_vs_3s6z | **83.7±2.7** | 62.4±40.4 | 56.5±23.0 | 4.0±7.0 | 32.7±28.3 |
| MMM2 | **84.0±7.9** | 74.0±6.2 | 77.5±15.2 | 79.3±9.2 | 73.0±1.4 |

ences $\{o_t, u_t, o_{t+1}, r_t\}$ can be sampled randomly regardless of the agent. The total loss for the actor is defined as follows

$$L(\phi) = \frac{1}{NT} \sum_{n=1}^{N} \sum_{t=0}^{T-1} \left[ L_{n,t}^{\text{act}}(\phi) + \alpha L_{n,t}^{\text{pred}}(\phi) \right] \tag{6}$$

where $\alpha$ is the weight coefficient for self-predictive representation learning. We share the policy among all the agents, thus the agent index $n$ is omitted. The policy loss $L_t^{\text{act}}(\phi)$ and the self-predicting loss $L_t^{\text{pred}}(\phi)$ are given by

$$L_t^{\text{act}}(\phi) = \min \left[ \frac{\pi_\phi(u_t|o_t)}{\pi_\phi^{\text{old}}(u_t|o_t)} A_\psi^{\pi_\phi}(s_t, \boldsymbol{u}_t), \text{clip}\left( \frac{\pi_\phi(u_t|o_t)}{\pi_\phi^{\text{old}}(u_t|o_t)}, 1 - \eta, 1 + \eta \right) A_\psi^{\pi_\phi}(s_t, \boldsymbol{u}_t) \right] \tag{7a}$$

$$L_t^{\text{pred}}(\phi) = \max \left( \mu, \text{KL}\left[ \text{sg}(\text{softmax}(z_t)) || \text{softmax}(\hat{z}_t) \right] \right) \tag{7b}$$

where $\pi_\phi^{\text{old}}$ is the previous policy used in collecting the data $\{o_t, u_t, o_{t+1}, r_t\}$, $\pi_\phi$ is the current policy, $A_\psi^{\pi_\phi}(s_t, \boldsymbol{u}_t)$ is the advantage function, $\psi$ represents the learnable parameters of the critic function, $\eta$ is the clip parameter that controls the controls the deviation between the current policy $\pi_\phi$ and the old policy $\pi_\phi^{\text{old}}$, sg($\cdot$) is the operation of stop-gradients, and $\mu$ is the early-stop coefficient of self-predictive representation training. We compute $A_\psi^{\pi_\phi}$ utilizing the $K$-step general advantage estimation (GAE)

$$A_\psi^{\pi_\phi}(s_t, \boldsymbol{u}_t) = \delta_t + (\gamma\lambda)\delta_{t+1} + \cdots + (\gamma\lambda)^{K-1}\delta_{t+K-1} - V_\psi^{\pi_\phi}(s_t) \tag{8}$$

$$\text{where} \quad \delta_t = \gamma V_\psi^{\pi_\phi}(s_{t+1}) + r(s_t, \boldsymbol{u}_t) - V_\psi^{\pi_\phi}(s_t)$$

where $r(s_t, \boldsymbol{u}_t)$ is the shared team reward, and $V_\psi^{\pi_\phi}(s_t)$ is the critic function with learnable parameters $\psi$.

The self-predicting loss $L_t^{\text{pred}}(\phi)$ is the KL divergence between the discrete categorical distributions parameterized by the encoded observation $z_t$ and the predicted transition $\hat{z}_t$. We stop propagating the gradients of softmax($z_t$) in $L_t^{\text{pred}}(\phi)$. We also halt the training of the transition decoder early by setting $L_t^{\text{pred}}(\phi)$ to $\mu$ if the self-predicting loss falls below a constant $\mu$ to reduce its interference in policy training when the transition prediction is relatively accurate.

**Critic learning** The critic function $V_\psi^{\pi_\phi}(s_t)$, implemented as a gated recurrent unit (GRU), is trained via self-supervised manner. We choose the advantage function in equation 8 as the critic target and use the mean squared error (MSE) as the critic loss

$$L(\psi) = \frac{1}{T} \sum_{t=0}^{T-1} \text{MSE}[A_\psi^{\pi_\phi}(s_t, \boldsymbol{u}_t)] \tag{9}$$

## 5 EXPERIMENTS

We evaluate the performance of self-predictive Mamba policy on six of the most challenging SMAC tasks, featuring asymmetric force distributions and heterogeneous agents. Our experiments aim to address the following key questions:

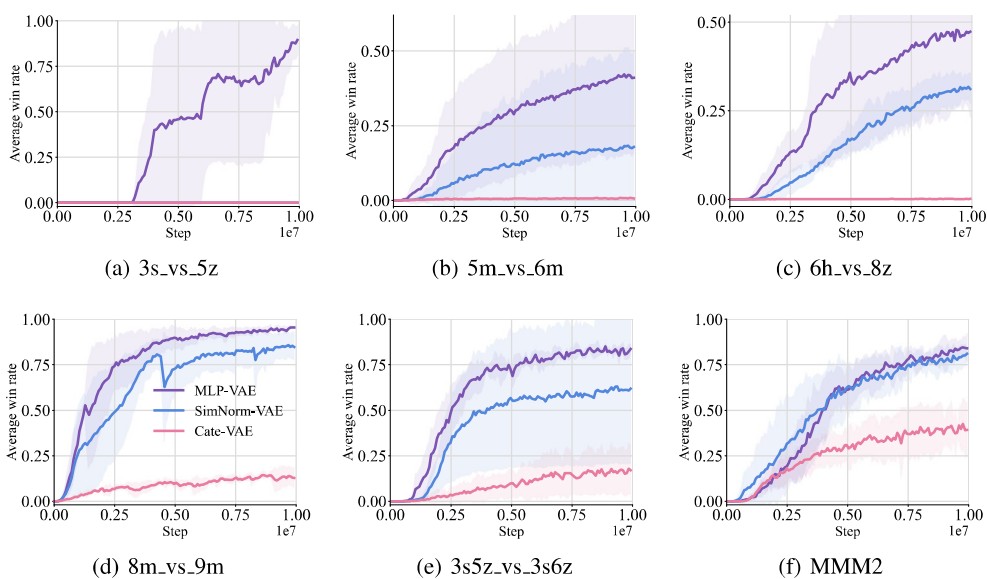

Figure 2: Comparison between different self-predictive Mamba polices using MLP-VAE, SimNorm-VAE and categorical-VAE.

- Q1. How effective is the self-predictive Mamba policy compared to a GRU-based policy? Additionally, how effective is self-predictive representation learning compared to directly using the Mamba model as the policy?

- Q2. Is the MLP-VAE structure effective compared to other proven effective architectures, specifically the categorical-VAE employed by Dreamer (Hafner et al., 2023) and the SimNorm-VAE employed by TDMPC2 (Hansen et al., 2024a)?

- Q3. Is the current design of self-predictive Mamba policy effective enough?

To address Q1, we conduct comparative evaluations among self-predictive Mamba policy, the naive Mamba policy, and the GRU-based policy. Details of these evaluations are provided in Section 5.2. To address Q2, we compare the MLP-VAE in self-predictive Mamba with the categorical-VAE from Dreamer and the SimNorm-VAE from TDMPC2. These comparisons are outlined in Section 5.3. To address Q3, we perform ablation studies that explore alternative structures designed for self-predictive Mamba policy, as described in Section 5.4.

## 5.1 EXPERIMENTAL SETTINGS

### 5.1.1 BENCHMARK

We conduct our experiments on SMAC, a benchmark including various multi-agent combat tasks that require defeating enemy force controlled by the built-in AI script in StarCraft II, which is well-known for its complexity and scalability. We carefully select six most challenging tasks in SMAC for evaluating the performance of the self-predictive Mamba policy against baselines, specifically 5m_vs_6m, 8m_vs_9m, 6h_vs_8z, 3s5z_vs_3s6z, MMM2 and 3s_vs_5z. All the selected tasks include asymmetric force distributions, implying that the enemy force is always stronger than the ally force. Specifically, 5m_vs_6m and 8m_vs_9m are basic asymmetric combat tasks with different scales, while 5m_vs_6m presents a greater challenge due to the higher force ratio between opposing sides; 8m_vs_9m is more difficult to learn because it involves a larger number of agents, and 3s5z_vs_3s6z and MMM2 are asymmetric tasks with multiple types of heterogeneous agents which significantly increases their complexity. Moreover, 3s_vs_5z is a task requiring long-term skill learning.

### 5.1.2 BASIC SETTINGS

We build self-predictive Mamba upon the implementation of MAPPO (Yu et al., 2022), but use the same hyperparameter settings among all the tasks and all the compared algorithms, except in 3s_vs_5z which requires long term skill learning, the historical observation sequence length is set to $k = 8$ and the early-stop coefficient of the self-predicting loss is set to $\epsilon = 0$. The maximum sampling step is set to 1e7. In comparison with baselines we add QMIX (Rashid et al., 2020) and its variant RODE (Wang et al., 2021) as reference. To evaluate the learning efficiency of the algorithms, we report the results of QMIX and RODE at the same runtime as self-predictive Mamba. We run all the experiments with 4 different random seeds in an environment with Python 3.7. We choose StarCraft II 2.4.6 for a more challenging task settings. All the comparisons are conducted on a PC with RTX3090. The full hyperparameter settings can be found in Appendices.

### 5.2 Q1. COMPARISONS WITH BASELINES

In this section, we present a detailed comparison of the self-predictive Mamba policy against two key baselines: the GRU-based policy, which is part of the original MAPPO implementation, as well as the vanilla Mamba policy, where the Mamba model is directly used as the policy network. As shown in Table 1, the self-predictive Mamba policy consistently outperforms both baselines across all the tasks evaluated. The full results among self-predictive Mamba, vanilla Mamba and GRU policy can be referred to Appendix B. Although the Mamba policy demonstrates a marginal improvement over the GRU-based policy, the performance gain is limited without the self-predictive representation learning module. In contrast, the self-predictive Mamba policy's significant performance advantage is attributed to its ability to learn self-predictive features by means of the novel MLP-VAE structure. This feature enhancement enables the self-predictive Mamba policy to make more informed and efficient decisions in these challenging multi-agent environments in SMAC. The results suggest that the self-predictive mechanism plays a crucial role in boosting the self-predictive Mamba policy's ability to handle asymmetric and heterogeneous agent tasks, confirming its effectiveness over the existing GRU-based and vanilla Mamba policies.

### 5.3 Q2. DESIGNING ENCODER STRUCTURE

#### 5.3.1 ALTERNATIVE STRUCTURES

Recently, the most popular VAE implementation in RL is categorical-VAE which samples a multiple categorical variable as the encoded features. The categorical encoding progress is described by

$$z_t \sim q_\phi(z_t|o_t) = \mathcal{Z}_t \tag{10}$$

where $\mathcal{Z}_t$ is a distribution comprising multiple categories. The latent variable $z_t$ is then sampled from $\mathcal{Z}_t$ to represent the raw input $o_t$. It was implemented in e.g., DreamerV3 (Hafner et al., 2023) and its various variants (Robine et al., 2023; Zhang et al., 2023). However, multiple categorical variables may lose too much information when handling non-image observations encountered in e.g., SMAC. Furthermore, Dreamer utilizes multiple categorical variables with $32 \times 32 = 1024$ dimensions, which are necessary for processing the image inputs in Atari, but it is overly redundant for compressing input features in SMAC.

Considering the characteristics of non-image observations, TDMPC2 (Hansen et al., 2024a) introduced the SimNorm-VAE, which encodes raw inputs into multiple categorical variables without performing any sampling. Let us start by defining $d = M^2$ for any $M \in \mathbb{N}_+$ and $\tilde{z}_t = f_{\text{MLP}}(o_t) \dot{=} [x_1, x_2, ..., x_d]_t \in \mathbb{R}^{1 \times d}$, the $\tilde{z}_t$ could be divided into $M$ partitions (groups) $[g_1, g_2, ..., g_M]_t$, the $i$-th partition $g_i$ contains $M$ elements $[x_{(i-1) \times M+1}, x_{(i-1) \times M+2}, ..., x_{i \times M}]_t$. Thus, the SimNorm encoding is held as follows

$$z_t \dot{=} [\tilde{g}_1, \tilde{g}_2, ..., \tilde{g}_M]_t, \ \tilde{g}_i = \frac{e^{x_{(i-1) \times M+j}/\tau}}{\sum_{j=1}^{j=M} e^{x_{(i-1) \times M+j}/\tau}} \tag{11}$$

where $\tau$ is the temperature parameter that modulates the sparsity of $\tilde{g}_i$. SimNorm-VAE shows competitive performance in TDMPC2 in continuous controlling tasks in DMControl (Tassa et al., 2018), Meta-World (Yu et al., 2020), ManiSkill2 (Gu et al., 2023), and MyoSuite (Caggiano et al., 2022) with non-image observations. Nevertheless, the softmax operation in equation 11 constrains the

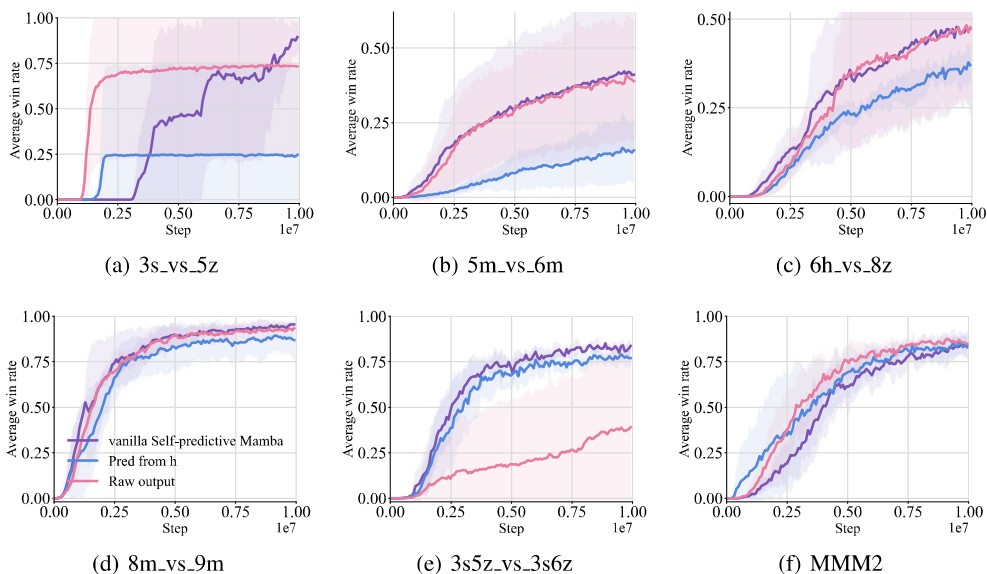

Figure 3: The details of the comparison among vanilla self-predictive Mamba and 2 alternative structures.

model's capacity for feature learning in the latent state, which may lead to suboptimal performance in SMAC, where observation inputs exhibit rapid and discontinuous changes.

Due to the described limitations of the aforementioned encoder structures, we advocate the MLP-VAE encoder in equation 4a tailored to the characteristics of agent observation inputs in the SMAC environment. Figure 2 shows the comparison among MLP-VAE, categorical-VAE and SimNorm-VAE. As the hidden state of the model is set to be 64 dimensions, the distribution $\mathcal{Z}_t$ of the categorical-VAE is set to include 8 categoricals, each includes 8 classes. The scale $M$ of the SimNorm-VAE is also set to 8. The temperature coefficient $\tau$ is set to 1.

The self-predictive Mamba utilizing categorical-VAE performs the worst due to the excessive information loss from discrete encoding. Moreover, the self-predicting loss curves in Appendix C demonstrates that the features encoded by the categorical-VAE even fails to correctly capture the self-predictive representations. The self-predictive Mamba with SimNorm-VAE outperformed the version with categorical-VAE in self-prediction learning, and this advantage is also reflected in the final performance. However, due to the softmax operation during encoding, it performed worse than the self-predictive Mamba with MLP-VAE. This highlights the superior encoding capability of the proposed MLP-VAE, which extracts the self-predictive representations efficiently from the raw inputs with greater accuracy and completeness, and thus lead to the strongest performance.

### 5.3.2 OBJECTIVES OF LEARNING

The most widely-used learning objectives for VAE variants includes minimizing the self-predicting loss $L^{\mathrm{pred}}(\phi)$ along with the representation loss $L^{\mathrm{rep}}$ and the reconstruction loss $L^{\mathrm{rec}}$

$$L_t^{\mathrm{rep}}(\phi) = \max\left(\mu, \mathrm{KL}\left[\mathrm{softmax}(z_t) || \mathrm{sg}(\mathrm{softmax}(\hat{z}_t))\right]\right) \tag{12a}$$

$$L_t^{\mathrm{rec}}(\phi) = \mathrm{MSE}[o_t - \mathrm{REC}_\phi(z_t)] \tag{12b}$$

where $\mathrm{REC}_\phi$ is a decoder for reconstructing the raw input $o_t$; $L^{\mathrm{rep}}(\phi)$ slightly aligns the encoder output with the predicted results, making the encoded representations more amenable to prediction, and $L^{\mathrm{rec}}(\phi)$ ensures that the encoder output retains as much of the original input's features as possible. The total VAE loss is given by

$$L^{\mathrm{VAE}}(\phi) = \alpha L^{\mathrm{pred}}(\phi) + \beta L^{\mathrm{rep}}(\phi) + \sigma L^{\mathrm{rec}}(\phi) \tag{13}$$

where $\alpha, \beta, \sigma$ are weight coefficients. Nonetheless, we suppose that in policy learning, the primary objective is to extract features highly relevant to decision-making. Preserving all input features may

introduce irrelevant information, which could hinder the policy learning performance. Besides, the self-predicting loss is designed for assisting the policy learning in this work, so it seems unnecessary to add the representation loss. We compare the loss function $L^{\text{SP-Mamba}}(\phi) = \alpha L^{\text{pred}}(\phi)$ designed

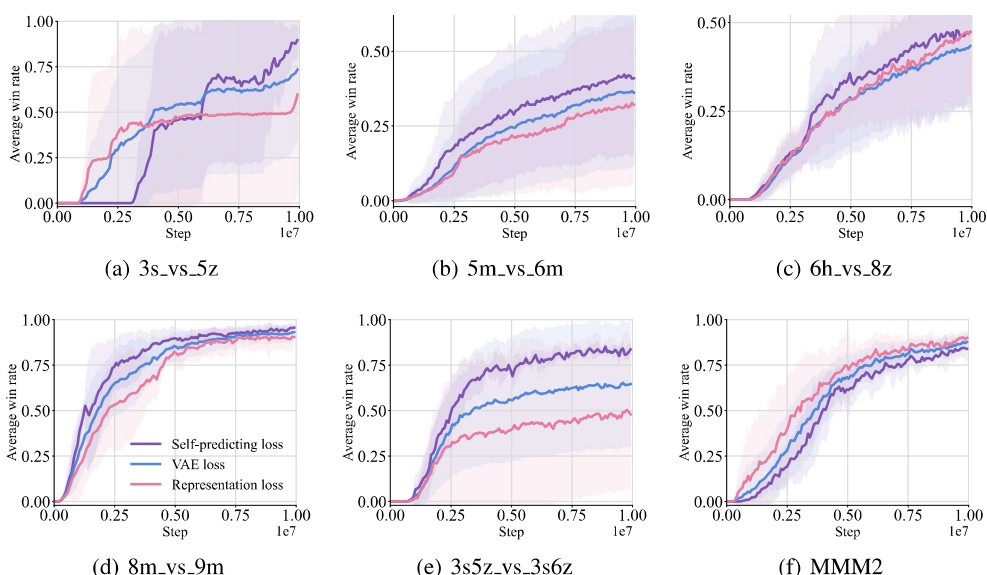

Figure 4: Comparison between the $L^{\text{SP-Mamba}}(\phi)$, $L^{\text{REP}}(\phi)$, and $L^{\text{VAE}}(\phi)$ losses.

for self-predictive Mamba against 2 alternative loss functions $L^{\text{REP}}(\phi) = \alpha L^{\text{pred}}(\phi) + \beta L^{\text{rep}}(\phi)$ and $L^{\text{VAE}}(\phi)$ in equation 13. The coefficients are set to $\alpha = 0.1$, $\beta = 0.02$, and $\sigma = 0.2$. The results can be found in Figure 4. Clearly, incorporating the reconstruction objective significantly degrades policy performance, indicating that learning fully reconstructable features interferes with decision-making. Meanwhile, the representation loss has minimal impact on the performance.

### 5.4 Q3. ABLATION STUDIES

As an advanced recurrent model, the self-predictive representation learning structure for Mamba should be designed carefully. We present two different self-predictive representation learning structures. The 'Pred from $h_t$' configuration makes the transition decoder predict the next output of the encoder from Mamba's hidden state $\hat{z}_{t+1} = \text{DEC}_\phi(h_t, u_t)$, and the transition decoder in 'Raw output' configuration predicts the next output from $y_t$ that is not been normalized. The results in Figure 3 indicate that neither predicting from $h_t$ nor from the un-normalized $y_t$ is beneficial to policy learning. Since Mamba expands the dimensionality of $h_t$ by a factor of $\epsilon$ relative to the input-output, directly predicting $\hat{z}_{t+1}$ from $h_t$ may be inefficient. Yet, predicting based on un-normalized features performs slightly worse, as the un-normalized features exhibit numerical instability.

### 6 CONCLUSION

In this work, we introduced self-predictive Mamba, a novel framework that integrates the Mamba model with self-predictive representation learning to improve decentralized policy optimization in multi-agent systems. The self-predictive Mamba utilizes a Mamba model as the core part of processing agents' historical observations, and trains the Mamba model to predict future observations. It achieves superior performance against traditional RNN-based policies in extensive experiments without careful manual hyperparameter tuning according to the difficulty of tasks. The success of self-predictive Mamba highlights the potential of the Mamba model as a powerful module for sequential decision-making tasks, offering a scalable and efficient approach for handling complex multi-agent environments.

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
