# OpenReview forum: "Self-predictive Mamba: Improving Multi-agent Reinforcement Learning with Self-predictive Encoding"
_ICLR.cc/2025/Conference — Submitted to ICLR 2025_

### Official Review · Reviewer_n4kh · 2024-11-01

**Soundness:** 2
**Presentation:** 2
**Contribution:** 2
**Rating:** 3
**Confidence:** 4

**Summary:**

This paper presents an approach to multi-agent reinforcement learning that aims to improve the handling of partial observability and non-stationarity in decentralized settings. The main focus is on using a Mamba-based architecture combined with self-predictive learning to better handle historical information in MARL. This is done via several steps. First, observations are encoded using an MLP-VAE to create a latent representation, chosen over categorical-VAE and SimNorm-VAE alternatives which they show perform worse. This representation is then processed through a Mamba latent model that updates a hidden state. The output is projected and fed to a decision maker that produces categorical action distributions. A transition decoder attempts to predict future encoder outputs, providing a self-predictive learning signal. They integrate this with MAPPO for policy learning and demonstrate their approach on several SMAC tasks. They then conduct ablation studies comparing different loss functions, showing that reconstruction objectives can harm performance, and analyze different encoder architectural choices.

**Strengths:**

Letting the decoder predict the next encoder output instead of reconstructing the observation to learn better representations is interesting and something that can be investigated further.
Strong results on the SMAC environments which were used.
Reasonable robustness in the results that were shown. It is positive to see error bars shown, as this is often left out.
Some ablation studies included.

**Weaknesses:**

Unfortunately the authors do very little evaluation and only evaluate their method on 6 SMAC scenarios despite claiming that evaluation is extensive. The authors also claim that SMAC is very challenging, but literature has shown that the benchmark is saturated and overfit to [1], the the benchmark is trivial and decent policies can be learnt by only conditioning on agent identifiers and the current timestep [2] and that it is possible to attain nearly 100% on most scenarios used in this paper using only MAPPO [3].

In 4 out the 6 tasks tested, the performance reported in Table 1 does not match the performance of MAPPO (which the code for this work extends). And in 2 of the tasks SP-MAMBA overlaps with the result of MAPPO as reported. The QMIX baselines used are also worse than those reported in [3].

The chosen tasks are also not sufficiently difficult given that MAPPO can get 100% win rates on some of them.
Using an MLP as VAE does not seem novel to me.

[1] Gorsane, Rihab, et al. "Towards a standardised performance evaluation protocol for cooperative marl." Advances in Neural Information Processing Systems 35 (2022): 5510-5521
[2] Ellis, Benjamin, et al. "Smacv2: An improved benchmark for cooperative multi-agent reinforcement learning." Advances in Neural Information Processing Systems 36 (2024).
[3] Yu, Chao, et al. "The surprising effectiveness of PPO in cooperative, multi-agent games (2021)." arXiv preprint arXiv:2103.01955 (2021).

**Questions:**

The results are worse than those in the original MAPPO paper although the code is based on the code from that paper. Do the authors know why this is the case?
When shuffling all the data in the buffer and combining data into a single batch, is the time ordering maintained in the sequences?
It seems that the model has to keep a cached sequence of observations during inference, does this not impact the model’s inference time memory requirements?
Why do the authors maintain a hidden state and then also condition on a cached sequence of observations?

---

### Official Review · Reviewer_RK1D · 2024-11-01

**Soundness:** 1
**Presentation:** 3
**Contribution:** 2
**Rating:** 3
**Confidence:** 4

**Summary:**

This paper proposes a method to use Mamba as a multi-agent reinforcement learning (MARL) policy. The proposed method is called SP-Mamba. The standard method for allowing policies to memorise previous states is to use recurrent neural networks (RNNs) however, this work proposed Mamba as it has been shown to outperform RNNs in other tasks. The authors show that through a self-predictive loss, where the policy predicts the next encoded observation, they are able to stabilise the learning of SP-Mamba and outperform both strong baselines and a naive mamba implementation (without the self-predictive loss). Additionally, ablations are performed to find the best variational autoencoder and the best combination of objectives to train the VAE’s self-predictive loss.

**Strengths:**

The idea to use Mamba with a self-predictive loss is a novel approach to stabilising Mamba in MARL. Additionally, the way the self-predictive loss is constructed by focusing on reward maximisation rather than reconstruction, as is common in world model training, is a useful insight. Finally, the ablations performed are thorough and well-considered as they ablate useful features of the architecture.

**Weaknesses:**

The main issue I have with this work is that the claims made throughout the paper do not align with the results.
* In the introduction, the authors mention “Substantial experiments demonstrate…”. Substantial experiments were not performed. Only six scenarios were tested, from a single environment suite (SMAC). This is not considered substantial and is in fact around the average for a MARL paper [1].
* In the conclusion, they say “scalable and efficient approach for handling complex multi-agent environments”. It is not explained how this work is scalable and in what dimension it can scale e.g. number of agents, timesteps or model size. Additionally, it should not be claimed that this work can handle complex MARL environments as it is only tested on 6 tasks from a single environment suite, from these results, one cannot assume it will generalise to arbitrary complex environments.

Additionally, I find the following to be weaknesses of this work
* It seems suspect that the state-of-the-art (SOTA) method in cooperative MARL is referenced in this work - Multi-Agent Transformer [6] - however it is not used as a baseline. It would make sense to include this not only because it is SOTA, but also because it is a transformer-based policy, which is more similar to Mamba than the RNN-based policies used in recurrent PPO, QMIX and RODE. MAT also tests on SMAC and indeed on all the same tasks used in this work and in all cases MAT significantly outperforms the results reported in this work.
* To expand on this point, the results do not seem to align with previous work which has tested PPO and QMIX and in most cases significantly underperforms the results from multiple other independent works [4,5,6].
* There seems to be a lack of hyperparameter tuning which may significantly affect the results. This is cited as a strength in the conclusion of the paper, but I do not think it is reasonable to expect the hyperparameters that work well for SP-Mamba to also perform well in PPO. Additionally, it is not clear how hyperparameters were chosen. Was there some empirical evidence to support the choice of the hyperparameters selection? If so, this should be included in the paper.
* It is concerning that this work only tests on 6 tasks from a single suite. Gorsane et al. [1] recommend at least two distinct environment suites. This calls the significance of the results into question. Additionally [1] and [2] recommend evaluating with at least 10 seeds, this work only evaluates using 4 seeds.
* It is no longer recommended to test on SMAC v1 as it suffers from a lack of stochasticity and partial observability [3]. Instead, SMAC v2 [3] should be used.
* Given that Mamba should improve the memory capabilities of the approach it would have been interesting to see how it performs in more memory-intensive environments.


References:

[1] Gorsane, R., Mahjoub, O., de Kock, R.J., Dubb, R., Singh, S. and Pretorius, A., 2022. Towards a standardised performance evaluation protocol for cooperative marl. Advances in Neural Information Processing Systems, 35, pp.5510-5521.

[2] Agarwal, R., Schwarzer, M., Castro, P.S., Courville, A.C. and Bellemare, M., 2021. Deep reinforcement learning at the edge of the statistical precipice. Advances in neural information processing systems, 34, pp.29304-29320.

[3] Ellis, B., Cook, J., Moalla, S., Samvelyan, M., Sun, M., Mahajan, A., Foerster, J. and Whiteson, S., 2024. Smacv2: An improved benchmark for cooperative multi-agent reinforcement learning. Advances in Neural Information Processing Systems, 36.

[4] Rashid, T., Samvelyan, M., De Witt, C.S., Farquhar, G., Foerster, J. and Whiteson, S., 2020. Monotonic value function factorisation for deep multi-agent reinforcement learning. Journal of Machine Learning Research, 21(178), pp.1-51.

[5] Yu, C., Velu, A., Vinitsky, E., Gao, J., Wang, Y., Bayen, A. and Wu, Y., 2022. The surprising effectiveness of ppo in cooperative multi-agent games. Advances in Neural Information Processing Systems, 35, pp.24611-24624.

[6] Wen, M., Kuba, J., Lin, R., Zhang, W., Wen, Y., Wang, J. and Yang, Y., 2022. Multi-agent reinforcement learning is a sequence modeling problem. Advances in Neural Information Processing Systems, 35, pp.16509-16521.

**Questions:**

* On line 53 the authors say that MAT has stringent assumptions, what are these assumptions?

* On line 61 it is mentioned that SSMs are difficult to train for decision-making tasks, why is S5 [7] not discussed, where an SSM is used for partially observable single-agent RL tasks?

* There is a mistake on line 92, a should be u

* Why is the critic not trained using Mamba also, it seems strange that a GRU is used, but it is not discussed why in a paper about using Mamba instead of a GRU.

* Q3 on line 301/302 is vague. Effective in terms of what metrics and in comparison to which baseline?

* On line 363-365 you discuss that multiple categorical variables with a size of 32 x 32 will lose too much information, but were smaller sizes considered and experimented with?

* On line 406-409 the hyperparameters of the various VAEs are discussed, how were these chosen, are they standard values from previous work?

References:

[7] Lu, C., Schroecker, Y., Gu, A., Parisotto, E., Foerster, J., Singh, S. and Behbahani, F., 2024. Structured state space models for in-context reinforcement learning. Advances in Neural Information Processing Systems, 36.

---

### Official Review · Reviewer_WZ2m · 2024-11-04

**Soundness:** 2
**Presentation:** 3
**Contribution:** 2
**Rating:** 3
**Confidence:** 4

**Summary:**

This paper introduces self-predictive Mamba for decentralized policy optimization in multi-agent reinforcement learning (MARL). The authors aim to address the challenge of partial observability and non-stationarity in MARL by leveraging self-predictive representations. Specifically, the authors proposed (1) using MAMBA as the policy architecture, (2) using  MLP-VAE, which aggregates dense representations from raw observations, and (3) using the MAMBA model to predict feature encoded observations. The proposed approach is evaluated on six SMAC tasks. The experimental results show that self-predictive Mamba outperforms RNN-based MARL policies and those naively employing the Mamba model.

**Strengths:**

**Originality & Significance**.

The proposed approach merges MAMBA with self-predictive RL. While this combination may seem straightforward, given the success of both MAMBA and self-predictive RL, the reviewer believes the topic still holds significant interest for our community. It could provide valuable insights into the application of MAMBA within the realm of reinforcement learning. Also, the approach is well motivated and well-placed in the literature.



**Clarity**

This well-organized paper is generally easy to follow, though some of the technical content could be clarified further. The writing is overall clear.

**Weaknesses:**

The reviewer's primary concern centers on the experimental results. The paper lacks clarity regarding the settings for each method, such as the number of training steps and overall training time, making a thorough comparison challenging. Notably, the reported win rates for the baseline methods (RODE, QMIX) appear significantly lower than those documented in existing literature. While the authors mention in the captions that the results are based on equivalent training time, they offer no further explanation or justification. Sample efficiency is a critical aspect of evaluating RL methods. The reviewer questions why the win rate is not reported against the number of environment steps, as this omission hinders drawing definitive conclusions about the method's performance. Additionally, the caption of Table 1 references runtime data in Appendix B, but the reviewer could not locate this information.


Furthermore, Table 1 only includes a comparison with two baseline methods. To strengthen the evaluation, the authors should consider comparing their approach with more baselines like FT-QMIX [a], Qplex [b], and MAPPO [c] across a wider range of tasks. Given that the proposed method is built upon MAPPO, a direct comparison with this baseline is particularly crucial for assessing the effectiveness of the novel approach.


[a] Rethinking the Implementation Tricks and Monotonicity Constraint in Cooperative Multi-Agent Reinforcement Learning, Hu 2023.

[b] QPLEX: Duplex Dueling Multi-Agent Q-Learning, Wang 2020.

[c] The Surprising Effectiveness of PPO in Cooperative, Multi-Agent Games, Yu 2022.

**Questions:**

1. How do the authors select these six tasks? Could you consider including additional tasks, such as corridor or 10m vs. 11m?

2. Reporting the win rate versus environment steps is crucial.

---

> ### Comment · Reviewer_WZ2m · 2024-11-27
>
> I appreciate the author's response, but my main concern is still the lack of a plot showing the win rate against the number of environment steps. This issue was not addressed in the response.

---

### Meta-Review · Area_Chair_qbqf · 2024-12-21

**Metareview:**

This paper addresses partial observability and nonstationarity in multi-agent setups by incorporating the Mamba model into policy learning, which is additionally learned through self-prediction objectives, called self-predictive Mamba. The authors claimed that a careful choice of the objectives and architecture led to stable learning and superior performance.

Can you make the PDF searchable?

In Figure 1, consider adding an input arrow from x_t for the output projection.

Should the y_t on the right-hand side of (5) be h_t?

Please describe explicitly how C_t of (1b) is related to (5).

If z hat of (4e) is supposed to be a prediction of encoding z, it is a bit problematic that z hat is introduced before z.

While you noted that the loss in (9) uses the advantage function defined in (8), there is a nuance here that must be clarified. The value estimate in (8) appears many times, evaluated at different values, each of which in (8) are written as a function of psi. When you calculate the gradient of (9) wrt psi, do you differentiate all appearances of the value estimate or only one of them? The latter would be a semi-gradient update and is what is typically used, including in PPO.

Mention near (13) that L^Pred is defined in (7b).

I believe this work has the potential to be an excellent contribution in the future by addressing some of the critical concerns brought up by the reviewers summarized below.

**Additional Comments On Reviewer Discussion:**

The reviewers unanimously agreed that the paper does not substantiate the empirical claims made in the paper. For example, reviewers pointed out that without sufficient detail about the number of training steps and total training time, the comparison against baselines might not be fair. Moreover, not using a state-of-the-art baseline such as Multi-Agent transformers is also mentioned. Using only a single task suite was also brought up as a concern.

---

### Decision · Program_Chairs · 2025-01-22

Reject